# The Impact of Free and Added Sugars on Cognitive Function: A Systematic Review and Meta-Analysis

**DOI:** 10.3390/nu16010075

**Published:** 2023-12-25

**Authors:** Kerri M. Gillespie, Melanie J. White, Eva Kemps, Halim Moore, Alexander Dymond, Selena E. Bartlett

**Affiliations:** 1School of Clinical Sciences, Faculty of Health, Queensland University of Technology, Kelvin Grove, QLD 4059, Australia; selena.bartlett@qut.edu.au; 2School of Psychology and Counselling, Faculty of Health, Queensland University of Technology, Kelvin Grove, QLD 4059, Australia; melanie.white@qut.edu.au; 3College of Education, Psychology and Social Work, Flinders University, Bedford Park, SA 5042, Australia; eva.kemps@flinders.edu.au; 4Laboratory of the Metabolic Adaptations to Exercise under Physiological and Pathological Conditions (AME2P), Université Clermont Auvergne, 63170 Clermont-Ferrand, France; halim.moore@connect.qut.edu.au; 5Mental Health and Specialist Services, Gold Coast Hospital and Health Service, Gold Coast, QLD 4215, Australia; alex.dymond@health.qld.gov.au

**Keywords:** free sugar, added sugar, glucose, fructose, SSB, sugar-sweetened beverage, HFCS, high-fructose corn syrup, cognition, executive function

## Abstract

A relationship between excessive sugar consumption and cognitive function has been described in animal models, but the specific effects of sugars in humans remains unclear. This systematic review and meta-analysis aimed to evaluate the current knowledge, research characteristics, and quality of evidence of studies investigating the impacts of free and added sugars on human cognition in healthy participants. The review identified 77 studies (65 experimental trials, *n* = 3831; 9 cross-sectional studies, *n* = 11,456; and 3 cohort studies, *n* = 2059). All cohort studies and eight of the nine cross-sectional studies found significant positive correlations between added sugar consumption and risk of cognitive impairment. Four studies identified reduced risk of cognitive impairment associated with natural fructose-containing foods. The majority of randomised control trials assessed short-term glucose facilitation effects on cognitive outcomes. The results from these studies suggest the need for a tightly regulated blood glucose level, dependent on individualised physiological factors, for optimal cognitive function. A meta-analysis of a subset of studies that assessed the impact of glucose on recall found improvements in immediate free recall compared to controls (*p* = 0.002). The findings highlight the potentially detrimental effect of excessive, long-term, or prenatal added sugar consumption on cognitive function. Further research is needed to examine the specific effects of free and added sugars on cognitive function.

## 1. Introduction

The Western diet, defined as one high in added sugar, fat, and salt, has been linked to neurocognitive dysfunction and to a number of diseases that are themselves associated with cognitive impairment (obesity, cardiovascular disease, diabetes, and depression) [1,2,3,4,5,6,7,8]. While the Western diet is understood to have deleterious effects on physical, psychiatric, and neurophysiological function, the proportion of risk of disease or impairment attributed to any individual macronutrients is still debated [9,10]. A growing body of evidence from human and animal studies suggests that free and added sugars by themselves may be a significant modifiable risk factor for cognitive impairment [9,11,12,13,14]. Free sugars refer to sugars that are found in honey, syrups, and juices or added in food preparation or manufacturing, such as glucose, fructose, sucrose (a sugar molecule made from glucose and fructose combined), and hydrogenated starch hydrolysates (high-fructose corn syrup) [15,16]. Added sugars, according to the European Food Safety Authority (EFSA), refer only to those sugars included in the food preparation and manufacturing process and do not include natural sugars found in fruits and fruit juices [17].

While the overconsumption of sugars and carbohydrates has been implicated in several disease states, sugar is essential for cognitive function. The monosaccharide, glucose, is the primary energy source for the mammalian brain, which requires about 20% of the glucose-derived energy provided by basal metabolism [18,19]. Consistent and tightly regulated glucose metabolism is required for neuronal function, ATP generation, cellular maintenance, and the synthesis of neurotransmitters [18,19]. Thanks to glucose homeostatic mechanisms, the brain is resilient to minor changes in blood glucose levels, but moderate or severe hypo- or hyperglycaemia can impair neuronal functioning and lead to cell death [20,21,22]. Recurrent episodes of hypo- or hyperglycaemic states are associated with increased risk of cognitive impairment and dementia [23,24].

Evidence in animal models indicates that overconsumption of free sugars can lead to molecular changes and cognitive impairment [25], particularly in hippocampal-dependent memory [12,26]. Long-term cohort studies and cross-sectional studies in humans have found significant associations between the consumption of added sugars, specifically sugar sweetened beverages (SSBs), and reduced cognitive function, poor memory performance, and higher risk of cognitive impairment [27,28].

A large number of studies conducted over the past 30 years have observed a transient facilitation in cognitive function after the consumption of glucose, named the glucose facilitation effect [29]. These studies typically administer 25 to 50 g of glucose with a saccharine or aspartame control, matched for sweetness and mouthfeel using artificial sweeteners or lemon juice. A wide range of cognitive tests of memory or problem solving are generally conducted, usually on a participant group that has fasted between 2 and 12 h. Improvements in cognitive function subsequent to glucose facilitation are most observable in older participants, suggesting a mediating effect of age, possibly via age-related beta-cell dysfunction [30,31]. Some studies have observed similar or improved facilitation effects in fasted participants after the consumption of other macronutrients, such as proteins or fats, suggesting a negative cognitive effect of fasting rather than a positive effect of glucose consumption [32,33]. Another concern with many of these experiments is the use of artificial sweeteners as a control group and often as a mixer for the treatment drink itself. Artificial sweeteners, such as aspartame and saccharin, have been linked to a number of cognitive and behavioural impairments [25], making it a potential confounder in cognitive testing. The reliability and validity concerns mean that the independent impact of added sugars on cognitive function remains unclear.

Previous systematic reviews have found inconclusive evidence to determine the impact of free sugars. A systematic review by Garcia and colleagues [34] on the effects of glucose and sucrose in experimental trials found only limited evidence to suggest any benefit from such sugars on cognitive function and found that differences in study characteristics and assessments made comparisons difficult. A meta-analysis by Sun and colleagues [35] found insufficient evidence to definitively implicate SSB consumption in cognitive dysfunction. However, this analysis was conducted on only 10 cross-sectional and cohort studies in older adults. To date, there have been no comprehensive systematic reviews on the impact of all sugar types on cognitive function in healthy humans, including both observational and interventional studies. Accordingly, the current review aimed to investigate and report the current evidence on the effects of sugars on cognitive function. We also evaluated the characteristics of research that has been conducted and present directions for future research.

## 2. Materials and Methods

This systematic review and meta-analysis was conducted according to the 2020 PRISMA reporting guidelines for systematic reviews and meta-analyses [36]. The review protocol is registered on the PROSPERO database (registration no. CRD42022347984).

### 2.1. Data

The article search was conducted in PubMed, Cinahl, Embase, Psych Articles, and PsycInfo from conception to July 2022. An updated search was conducted in July 2023. No new studies were identified.

### 2.2. Search Strategy

The following search terms were used: [(cognit* OR executive function* OR brain function* OR memory OR decision making OR inhibitory control OR mental planning OR psychomotor speed OR psycho-motor speed) AND (sugar* OR fructose OR sucrose OR glucose OR carbohydrate*) NOT diabet*]. Where applicable, the MeSH terms “cognition”, “executive function”, and “sugars” were included. The search and subsequent screening excluded all papers relating to psychiatric or physiological health conditions such as diabetes, cancer, stroke, and schizophrenia (see Appendix A for full search criteria). Only human studies are included in the present paper. The search was limited to studies published in English. Citation chaining of identified papers and of systematic reviews and meta-analyses was used to identify studies that may have been missed.

### 2.3. Eligibility Criteria

Human studies of any age, location, study design, gender, or sample size were included. Studies had to be of primary data, investigate the effects of free or added sugars (glucose, sucrose, fructose, SSBs, or refined carbohydrates), and include a measure of cognitive function (such as memory, attention, processing speed, or inhibitory control). Studies were only selected if they included a control (comparisons between or within subjects) or placebo (in experimental trials) and a measure of sugars independent of other substances. Case reports, dissertations, reviews, and conference abstracts were excluded. Secondary use of data collected in national surveys, such as the National Health and Nutrition Examination Survey (NHANES), the Chinese Longitudinal Health Longevity Survey (CLHLS), and the Framingham study were excluded to avoid potential sampling bias and duplication of results.

### 2.4. Data Extraction

One investigator conducted the literature searches and imported all studies into the COVIDENCE systematic review management web application. Title and abstract screening was conducted by all authors. Two authors independently conducted full-text screening. Conflicts were resolved by discussion with a third researcher.

### 2.5. Quality Assessment

Two investigators independently assessed the risk of bias using the Cochrane Review Manager tool (RevMan, version 5.8.0). This online tool includes domains in random sequence generation, blinding, incomplete outcome data, and selective reporting. Each domain is judged to be high risk, unclear risk, or low risk (see Appendix A).

### 2.6. Analysis of Cognitive Function

A random-effects model meta-analysis was conducted for studies that included a measure of immediate and/or free recall with two comparator groups (placebo and glucose). Free recall was chosen as it was the most commonly used measure of cognition in all identified studies and therefore likely to obtain the largest number of homogenous data. To avoid bias from unequal comparisons, only studies administering 25–50 g of glucose were included. This was the most used dose of glucose in experimental studies. Meta-analyses were conducted using RevMan. Egger’s test for small study bias was conducted in SPSS version 29. An influence sensitivity analysis was performed by removing one study at a time to examine the effect of the excluded study on the pooled standard mean differences (SMD).

## 3. Results

### 3.1. Study Selection

A flow diagram describing the steps conducted during the process of the literature search and screening is presented in Figure 1. The search obtained 37,799 articles. After removal of duplicates, 27,030 articles underwent title and abstract screening, leaving 183 for full-text review. One of these articles was unable to be located. Full-text screening identified 74 eligible studies. A search of reference lists uncovered three new articles, resulting in a total of 77 studies eligible for analysis.

### 3.2. Study Characteristics

Included in the 77 eligible studies were 65 intervention studies, 9 cross-sectional studies, and 3 cohort studies. The smallest study included 11 participants, and the largest included 6387. Studies included a range of ages, with five studies comparing more than one age group. The 82 individual age groups investigated included children and adolescents (aged 5 to 18 years; *n* = 13), young adults (aged 18 to 37 years; however, one study included those aged 16 to 24; *n* = 41), mixed-age adults (18 years and over; *n* = 11), middle-aged adults (aged 35–55 years; *n* = 2), older adults (aged 55 years and over; *n* =11), mother–infant pairs (*n* = 2), and age not reported (*n* = 2).

### 3.3. Assessment of Study Quality

No studies were excluded from the analysis based on study quality. For the evaluation of RCTs, the reviewers assessed risk based on a per-protocol analysis of the data (that is, that the effects of the intervention are of primary interest rather than the effects of adherence to the intervention). Potential bias was most commonly due to a failure to adequately describe the process of randomisation, allocation concealment, or researcher and assessor blinding. Sample characteristics were often poorly described, and potential confounding factors were overlooked or not accounted for in analytic procedures. Only seven studies included a power calculation (included as “other bias”), which was considered significant for potential bias considering the small sample sizes of the included experimental studies (see Appendix A).

### 3.4. Cognitive Outcomes and Measures

A total of 86 different cognitive tests were used in the 77 studies. The most commonly used test was a word list test of auditory memory (recall and recognition) (*n* = 20), including the California Verbal Learning Test (CVLT-II) and the Rey Auditory Verbal Learning Test (RAVLT). Written word list tests of verbal memory were the second most commonly used test (*n* = 15). These were followed by digit span (*n* = 9), the STROOP test (*n* = 8), serial sevens or serial threes (*n* = 8), trail making tests (TMT) (*n* = 6), and choice reaction time tasks (*n* = 5). For ease of interpretation, findings of all tests have been categorised into the following domains of cognitive function: attention, memory (including declarative and working memory), executive function (including verbal and figural fluency, reasoning, shifting, planning, and problem solving), coordination (fine motor skills and hand–eye coordination), processing speed, perception, and global cognitive function.

### 3.5. Observational—Cross-Sectional Studies

The findings included nine cross-sectional studies published between 1982 and 2022 (see Table 1 for full details). Studies were conducted in the USA (*n* = 3) [38,39,40], China (*n* = 1) [41], Tibet (*n* = 1) [42], Finland (*n* = 1) [43], Kuwait (*n* = 1) [44], Malaysia (*n* = 1) [14], and Norway (*n* = 1) [45]. The sample size of the studies ranged from 54 to 6387 participants. Studies included eight investigations of children and adolescents and one of older adults. All nine studies used a food frequency questionnaire to collect dietary information. Cross-sectional studies investigated global cognitive function (*n* = 5) [14,38,39,43,44], memory (*n* = 4) [14,40,42,44], executive function (*n* = 3) [41,42,45], processing speed (*n* = 2) [14,42], and attention (*n* = 2) [14,42].

SSBs were investigated in five studies [14,41,42,44,45], and all were associated with impairment in executive functioning [41,42,45], global cognitive function [14,44], memory [42], and attention [42]. Glucose and sucrose were both tested in two studies [14,43] but were associated with impairment in global cognitive function in only one study of older adults [14]. Refined carbohydrates and added sugars were associated with impairment in global cognitive function [14,38,39] and memory [14] in three of four studies. Fructose (total dietary and natural fructose from fruit and fruit juice) was investigated in three studies [14,43,45]. Fruit and fruit juices were associated with superior performance on tests of global cognitive function [14,45] and executive function [45]. Total dietary fructose was associated with enhancements in global cognitive function in children and adolescents [43] but impairment in older adults [14].

### 3.6. Observational—Cohort Studies

The three cohort studies retrieved included one retrospective cohort study of mixed-age adults [48] and two prospective cohort studies of mother–infant pairs (*n* = 2) [46,47]. These were conducted between 2011 and 2020 in the USA. Two of the cohort studies followed mother–infant pairs. Berger [46] followed 88 mother–infant pairs over the first 24 months after birth and found that maternal consumption of fructose and SSBs (measured one month postnatally) was negatively associated with infant cognitive development at 24 months of age. Cohen [47] followed 1234 mother–child pairs for approximately 8 years, finding that maternal sucrose and SSB consumption, but not fructose or juice consumption, was negatively associated with cognitive function of the infant in mid-childhood. Their findings also showed that early childhood SSB consumption was associated with lower mid-childhood cognitive function but that fruit and fructose consumption was positively associated with early childhood vocabulary. The third large cohort study of 737 adults also found impairment in global cognitive function associated with fructose and SSB consumption [48]. Global cognitive function impairment was also associated with higher consumption of sucrose, glucose, and added sugars. Added sugars were also associated with poor memory [48].

### 3.7. Intervention Studies

The results of 65 intervention studies published between 1983 and 2020 were included. Studies were conducted in the UK (*n* = 42), the USA (*n* = 12), Australia (*n* = 5), Spain (*n* = 2), New Zealand (*n* = 1), Canada (*n* = 1), France (*n* = 1), and the Netherlands (*n* = 1). See Table 2 for full details. Studies included between 11 and 188 participants and were all randomised controlled trials (RCT), except for one nonrandomised, placebo-controlled trial [49]. All intervention studies investigated the short-term impact of sugars on cognitive function, from immediately postconsumption up to 24 h postconsumption (*n* = 1) [50]. Most studies tested cognitive function 10–15 min after consumption of the sugar or placebo treatment. A total of 9 of the 65 studies included numerous ages or sugar types, leaving 70 individual age groups (young adults (*n* = 41), all-age adults (*n* = 11), older adults (*n* = 10), children and adolescents (*n* = 5), middle-aged adults (*n* = 2), and age not described (*n* = 1)) and 74 individual sugar groups (glucose (*n* = 62), sucrose (*n* = 9), fructose (*n* = 2), SSB (*n* = 1)).

Artificial sugar was most commonly used as a control (*n* = 54); two studies used water [51,52], and nine did not disclose their control substance [53,54,55,56,57,58,59,60,61]. Artificial sugars used were saccharin (*n* = 26), aspartame (*n* = 17), aspartame and acesulfame K (*n* = 4), aspartame and saccharin (*n* = 3), sucralose (*n* = 2), xylitol (*n* = 1), or a mix of saccharides and stevia (*n* = 1). Nearly all studies required participants to fast for 8 h or more (*n* = 39) or between 2 and 4.5 h (*n* = 15) before testing. Nine studies did not enforce dietary restrictions, and two studies did not disclose whether participants were fasted or not. Studies that did not fast participants had a higher proportion of nonsignificant (33%) and impaired (22%) results after sugar consumption compared to the fasted groups (20% and 5%, respectively). However, there were not enough nonfasted studies to investigate the statistical significance of this relationship.

Experimental studies were most likely to conduct tests of memory (*n* = 52). Of these, 22 found improvements in performance after sugar consumption, 2 saw impairments, and 6 found mixed results (see Table 3 for summary findings). Also measured were processing speed (*n* = 45) and attention (*n* = 42), executive function (*n* = 19), perception (*n* = 17), and coordination and fine motor skills (*n* = 8). Overall, 36 studies found improvements in one or more measures of cognitive function, 5 found impairments, and 9 found mixed results.

**Table 2 nutrients-16-00075-t002:** Intervention trial characteristics.

Author/s	Population	Cognitive Measures	Intervention/Control	Fasting	Major Findings
Adan and Serra-Grabulosa, 2010 [51]	*N* = 72 young adults (36 male) aged 18–25 (M = 21.07, SD = 1.70)	RAVLT, Purdue–Pegboard, Benton Judgement of Line Orientation Test (JoLO), WCST, California Computerized Assessment Package (CalCAP), digit span of WAIS	(1) 75 mg caffeine, (2) 75 mg glucose, (3) 75 mg caffeine + 75 mg glucose, (4) Water	8 h	Water performed worse than treatment groups (*p* = 0.026). Glucose performed better on Purdue pegboard assembly than placebo or caffeine (*p* = 0.039). No effect of treatment on reaction time, WAIS, WCST, and RAVLT.
Allen et al., 1996 [62]	*N* = 28 elderly adults (6 male) aged 61–87 (M = 73)	Rey–Osterrieth Complex Figure, Taylor Complex Figure, dichotic listening, TMT, verbal fluency, Boston naming test, Meier visual test, grooved pegboard, figural fluency	(1) 50 g glucose, (2) 23.7 mg saccharin	9 h	Glucose enhanced delayed recall and verbal and figure fluency (*p*s < 0.001). No effect of glucose on other measures. Poor glucose regulation associated with worse performance in dichotic listening (*p* < 0.005) and verbal fluency (*p* < 0.05).
Azari, 1993 [63]	*N* = 18 young male adults aged 19–25 (M = 21, SD = 1.65)	Word recall	(1) 30 g glucose + 350 mg aspartame, (2) 100 mg glucose, (3) 450 mg aspartame	4.5 h	No effect of glucose or BGL on memory.
Benton et al., 1987 [64]	*N* = 60 children (30 male) aged 6 or 7 years old.	Paradigm of Shakow	(1) 25 g glucose, (2) Saccharin	2–3 h	Improved reaction time after glucose (*p* ≤ 0.05). Glucose group reported increased quiet concentration (*p* < 0.001) and less likely to fidget (*p* < 0.04).
Benton, 1990 [65]	T1: *N* = 20 male students (M age = 20.3 and 20.5 per group) T2: *N* = 40 undergraduate students (20 male; M age = 21.2 and 20.9)	T1: choice reaction time task T2: long arithmetic	(1) 25 g glucose, (2) Aspartame	4 h	T1: Glucose associated with fewer errors (*p* < 0.05). T2: No effect of glucose on coordination or arithmetic. Males performed better on arithmetic (*p* < 0.001) and coordination (*p* < 0.01).
Benton and Owens, 1993 [66]	T1: *N* = 153 undergraduate students (100 male, M age = 21.6, SD = 4.8; 53 female, M age = 21.8, SD = 5.2) T2: *N* = 53 female undergraduates (M = 21.5, SD = 5.0)	T1: word list, pattern recognition T2: word list, Weschler memory scale	T1: (1) 50 g glucose, (2) Aspartame + acesulfame K T2: (1) Sustained glucose (50 g + two 25 g top-ups)	No	T1: No effect of glucose on recall or spatial memory. Increased BGL associated with more words remembered (*p* < 0.002). T2: No effect of glucose on Weschler or word list. Falling BGL in glucose group associated with improved memory; falling BGL in placebo group associated with worse memory.
Benton et al., 1994 [67]	T1: *N* = 70 female undergraduates (M age = 21.46) T2: *N* = 50 male undergraduates (M age = 21.7)	T1: RIPT T2: STROOP	(1) 50 g glucose + 25 g glucose; (2) Aspartame + acesulfame K	No	Glucose associated with more errors than placebo in RIPT (*p* < 0.031). No effect of glucose on memory, reaction time, or STROOP.
Benton and Stevens, 2008 [68]	*N* = 16 children (7 male) aged 9 or 10 years	Recall of objects test, pattern recognition, paradigm of Shakow	(1) 25 g glucose, (2) 25 g xylitol	No	More pictures recalled after glucose consumption (*p* < 0.025). No effect of glucose on spatial memory.
Best et al., 2008 [69]	*N* = 45 adults (19 male) aged 40–63 (M = 52.1, SD = 5.9)	RAVLT, Self-Ordered Pointing Task (SOPT), digit span, WAIS matrix reasoning	(1) 25 g glucose, (2) 7 g saccharide + 2 drops stevia	2 h	No effect of glucose on any outcome.
Birnie et al., 2015 [70]	*N* = 16 adults (8 male) aged 18–45 (M = 23.7, SD = 5.0)	SART, Short Imaginal Processes Inventory (SIPI)	(1) 25 g glucose, (2) Saccharin	2 h	No effect of glucose on any outcome.
Brandt et al., 2006 [71]	T1: *N* = 40 undergraduate students (20 male) aged 18–25 T2: *N* = 40 undergraduate students (14 male) aged 18–36 (M = 22)	Word list	(1) 25 g glucose, (2) Aspartame	2 h	No effect of glucose on emotional memory enhancement. BGL were negatively associated with positive items remembered (*p* < 0.05). Better glucoregulation associated with improved memory for negative items and fewer errors (*p*s < 0.05).
Brandt et al., 2010 [72]	T1: *N* = 40 undergraduate students (5 male) aged 18–34 (M = 19.1) T2: *N* = 40 undergraduate students (27 male) aged 18–37 (M = 21)	Word list	T1: (1) 15 g glucose, (2) Aspartame T2: (1) 25 g glucose, (2) Aspartame	2 h	Greater false alarm rate after 25 g glucose condition (*p* < 0.05) Recognition memory was marginally improved in the aspartame group (*p* = 0.05).
Brandt et al., 2013 [73]	*N* = 60 undergraduate students (14 male; M age = 19.7)	STROOP	(1) 25 g glucose, (2) Aspartame	Overnight	Shorter reaction times in congruent and incongruent conditions after glucose consumption (*p*s < 0.05). Greater facilitative effect of glucose in incongruent (higher cognitive load) task.
Brandt, 2015 [74]	*N* = 41 undergraduate students (9 male; M age = 19.47)	Process dissociation procedure	(1) 25 g glucose, (2) Aspartame	Overnight	Exclusion (high effort condition) was superior in glucose condition (*p* < 0.05). No effect of treatment on inclusion. Aspartame scored higher in familiarity (*p* < 0.01) (low effort condition). No effect of treatment on recollection.
Brody and Wolitzky, 1983 [75]	*N* = 59 undergraduate students (28 male) aged 16–24 (M = 18.7)	Serial sevens	(1) 100 g sucrose, (2) 52 mg saccharin, (3) Water	8 h	No effect of treatment.
Brown and Riby, 2013 [76]	*N* = 35 young adults (14 male) aged 18–35 (M = 22.17, SD = 5.97)	Item recognition task, STROOP	(1) 25 g glucose, (2) 37.5 mg saccharin	2 h	Glucose facilitation effect in more difficult task conditions, but glucose impaired performance on new items (low cognitive load) (*p* = 0.02). No effect of glucose on attention.
Craft et al., 1994 [77]	*N* = 59 (27 younger adults aged 19–28, M = 20.8; 32 older adults aged 58–77, M = 68.5)	Paced Serial Addition Test, paragraph recall, modified CVLT, pattern recall and recognition, serial reaction time, word generation	(1) 50 g glucose, (2) 23.7 mg saccharin (a) Male vs. female, (b) Young vs. old, (c) Good vs. poor glucose control	Overnight	Glucose improved declarative memory in older males with good glycaemic recovery (*p* < 0.01). Glucose improved recall in younger men with poor glycaemic recovery (*p* < 0.01). Younger men with good glycaemic recovery saw memory deterioration after glucose consumption (*p* < 0.001). No effect of glucose on procedural memory, working memory, or verbal fluency.
Donohoe and Benton, 1999a [78]	T1: *N* = 67 female undergraduate students (M age = 21.8, SD = 5.1) T2: *N* = 69 female undergraduates (M age = 20.2, SD = 2.1)	T1: water jar test, Finding Embedded Figures Test, Baddeley Logical Reasoning Task T2: Controlled Oral Word Association Test, WAIS—Block design subtest, Porteus maze—adults/14 year olds	(1) 50 g glucose, (2) Aspartame and saccharin	No	T1: No effect of glucose on outcomes. T2: Improved verbal fluency after glucose consumption (*p* < 0.001). Faster time to solve Porteus maze for 14 year olds after glucose (*p* < 0.002). No effect of glucose on block design.
Donohoe and Benton, 1999b [79]	*N* = 188 female undergraduate students (M age 21, SD = 4)	T1: RIPT, word list	(1) 50 g glucose, (2) Aspartame and saccharin	No	Faster recall for glucose compared to control (*p* < 0.001). No effect of glucose on recall. More errors in the placebo group at 2, 4 (*p*s < 0.01), and 6 min (*p* < 0.05) (but not 8 or 10 min) of RIPT vigilance task.
Flint and Turek, 2003 [80]	*N* = 67 Undergraduate students (15 male) aged 18–50 (M = 19.49, SD = 4.35)	Test of Variable of Attention (TOVA)	(1) 10 mg/kg glucose, (2) 100 mg/kg glucose, (3) 500 mg/kg glucose, (4) 23.7 mg saccharin	8 h	100 mg/kg showed impaired impulsivity (greater postcommission response time variability) (*p* < 0.01). No effect of treatment on any other measure.
Ford et al., 2002 [81]	*N* = 20 undergraduate students aged 20–23	Tailored version of CDR Assessment Battery	(1) 25 g glucose, (2) 28 mg saccharin	Overnight	No effect of glucose on memory.
Foster et al, 1998 [52]	*N* = 30 female young adults aged 18–22 years (M = 19.5)	Modified CVLT, ROCF, digit span	(1) 25 g glucose, (2) 37.5 mg saccharin, (3) Water	9 h	Improved delayed recall after glucose consumption (*p* < 0.05). No effect of treatment on any other outcome.
Giles et al., 2018 [82]	*N* = 105 young adults (74 female; M age = 22.5, SD = 6.6)	Immediate and delayed recall, STROOP, N-back task, continuous performance task	(1) Sprite, (2) Sprite zero	12 h	Improved performance on sustained attention after sugar intake (*p* < 0.05). No effect of treatment on selective attention, verbal memory, or working memory.
Ginieis et al., 2018 [83]	*N* = 49 young adults (26 fasted (15 female, M age = 22.6, SD = 4.2) or 23 nonfasted (13 female, M age = 24.3, SD = 4.9))	Simple response time task, arithmetic task, STROOP	(1) 26 g glucose, (2) 14.5 g sucrose, (3) 13 g fructose, (4) 0.025 g sucralose, (5) Fasted vs. nonfasted	10 h	Slower reaction time after glucose consumption in the fasting group for simple response task and arithmetic task (*p* < 0.05). STROOP response time was impaired in the glucose and sucrose conditions, independent of fasting (*p* < 0.001).
Gonder-Frederick et al., 1987 [84]	*N* = 11 elderly adults aged 58–76 (M = 67.4, SD = 5.7)	WAIS memory subscales	(1) 50 g glucose, (2) 23.7 mg saccharin	9 h	Glucose group had improved performance on narrative memory (*p* = 0.024) and total Weschler scale (*p* = 0.009) (one-tailed). BGL at 30 and 60 min after beverage consumption was negatively associated with narrative memory, visual memory, and total Weschler scale (*p*s < 0.05).
Hope et al., 2013 [85]	T1: *N* = 12 young adults (6 male; M age = 25.1, SD = 2.1). T2: *N* = 24 young adults (3 male; M age = 20.1, SD = 0.7)	(1) Flanker task, simple version (2) Flanker task, demanding version	(1) 25 g glucose, (2) 2 mg saccharin	No	T1: Slower reaction time after glucose consumption (*p* = 0.03) only when glucose administered in session 1 and placebo in session 2. T2: Slower reaction time after glucose consumption (*p* = 0.045). No effect of glucose on error rates.
Jones et al., 2012 [32]	*N* = 18 young adults (5 male) aged 18–37 (M age = 19)	Tailored version of CDR Assessment Battery	(1) 40 g glucose, (2) 40 g protein + 2 g aspartame, (3) 16 g fat + 2 g aspartame, (4) 2 g aspartame	12 h	Enhancements in attention (*p* < 0.01) and speed (*p* < 0.05) 15 min after glucose ingestion. Impairments in working memory (*p* < 0.05) 60 min after glucose ingestion. Speed enhanced 15 min after fat consumption (*p* < 0.05). Working memory enhanced 15 min following protein ingestion (*p* < 0.05). Episodic memory and memory quality enhanced 60 min following protein ingestion (*p* < 0.01).
Kaplan et al., 2000 [31]	*N* = 20 older adults (10 male) aged 60–82	Tailored version of RAVLT, TMT, attention task (television recall)	(1) 50 g glucose, (2) 50 g carbohydrate from potato, (3) 50 g carbohydrate from pearled barley, (4) 23.7 mg saccharin	10–12 h	No effect of treatment on cognitive performance. Significant predictors of declarative memory were glycaemic regulation, BMI, and beta-cell function.
Kaplan et al., 2001 [33]	*N* = 22 older adults (11 male) aged 61–79 (M = 71.2, SD = 1.3)	Tailored version of RAVLT, paragraph recall, TMT, attention task (television recall)	(1) 50 g glucose, (2) 50.5 g whey isolate (protein) + 23.7 mg saccharin, (3) 41.4 g micro-lipid + 23.7 mg saccharin, (4) 23.7 mg saccharin	10–12 h	Protein, glucose, and fat (*p*s < 0.001) improved delayed recall at 15 min compared to placebo. Protein (*p* = 0.04), glucose (*p* = 0.02), and fat (*p* = 0.008) improved immediate recall at 15 min compared to placebo. No effect of treatment on recall or TMT at 60 min. Fat ingestion improved attention at 60 min.
Kennedy and Scholey, 2000 [86]	*N* = 20 young adults (6 male) aged 19–30 (M = 20.4)	Serial threes and sevens, word retrieval	(1) 25 g glucose, (2) 30 mg saccharin	9 h	Improved performance in serial sevens after glucose ingestion (*p* < 0.01). Performance in serial threes and sevens positively associated with fall in BGL during task (*p*s < 0.05). No significant effect of glucose on word retrieval.
Maben and Smith, 1996 [49]	*N* = 48 young adults (24 male) aged 18–32	Word list, Baddeley Logical Reasoning Task, Semantic memory task (not described)	(1) No sugar, (2) Sugar, (3) Aspartame, (4) No sugar + caffeine, (5) Sugar + caffeine, (6) Aspartame + caffeine	9 h	Sugar and aspartame conditions performed more accurately on logical reasoning than control (*p* < 0.05) but more slowly (*p* < 0.001). No effect of sugar or aspartame on free recall, semantic memory, or recognition memory.
Macpherson et al., 2015 [87]	*N* = 48 (24 young adults aged 18–23 (M = 20.6, SD = 1.4) and 24 older adults aged 65–85 (M = 72.5, SD = 5.1))	Auditory word recognition, target tracking task	(1) 25 g glucose (2) 30 mg saccharin	12 h	Tracking precision improved in older adults after glucose ingestion (*p* = 0.05) after controlling for BMI, IQ, and glucose regulation. No effect of glucose in younger adults.
Mantantzis et al., 2018 [88]	*N* = 112 (54 undergraduate students aged 18–27 and 58 older adults aged 65–82)	Choice reaction time task	(1) 25 g glucose (2) Aspartame	2 h	Glucose improved speed (*p* = 0.001) and accuracy (*p* = 0.007) in the older adult group only.
Martin and Benton, 1999 [89]	*N* = 80 female undergraduates (M age = 22.6)	Consonant trigrams	(1) 50 g glucose aft fast, (2) 50 g glucose after breakfast, (3) Aspartame + saccharin after fast, (4) Aspartame + saccharin after breakfast	Overnight fast vs. no fast	Performance improved over time for fasting plus glucose (*p* < 0.001), breakfast plus glucose (*p* < 0.03), and breakfast without glucose (*p* < 0.001). Falling BGL during the task was associated with better recall (*p* < 0.001).
Meikle et al., 2004 [53]	*N* = 25 adults (17 female) aged 18–52 (M = 28.4, SD = 9.3). Younger group (*N* = 14, M age = 21.8, SD = 3.3) and middle-aged group (*N* = 11, M age = 38.4, SD = 6.7)	Choice reaction time task, TMT, letter cancelation test, word retrieval, word list	(1) 25 g glucose, (2) 50 g glucose, (3) Placebo (4) Good vs. bad glucose regulators	9 h	Older adults saw improvement in reaction time for high memory load tasks after 25 g (*p* < 0.05) or 50 g (*p* < 0.01) of glucose. All participants had improved delayed free recall after 25 g (*p* < 0.05) and 50 g (*p* < 0.01) of glucose.
Meikle et al., 2005 [54]	T1: 37 young adults (29 female aged 17–48 (M = 28.3) T2: *N* = 24 young adults (20 female) aged 18–20 (M = 18.9)	Word list	T1: (1) 25 g glucose prelearning, (2) 25 g glucose postlearning, (3) Placebo T2: (1) 25 g glucose, (2) Placebo	9 h	T1: No main effect of treatment post hoc; for higher difficulty task, placebo forgot more words than glucose prelearning (*p* < 0.01) and postlearning groups (*p* < 0.05) T2: Glucose performed better on high memory load tasks (*p* < 0.01) but not on high cognitive demand tasks.
Messier et al., 1998 [90]	*N* = 100 female undergraduate students aged 17–48 (M = 21.3, SD = 4.6)	Word list	(1) 10 mg/kg glucose, (2) 100 mg/kg glucose, (3) 300 mg/kg glucose, (4) 500 mg/kg glucose, (5) 800 mg/kg glucose, (6) 1000 mg/kg glucose, (7) 52 mg saccharin, (8) Water	12 h	Increased primacy word recall after 10, 300, 500, 800, and 1000 mg/kg compared to one or both controls. Increased recency word performance after 500 mg/kg glucose. Impaired performance on recency word recall after 10 and 300 mg/kg glucose.
Miller et al., 2013 [91]	*N* = 36 adults (11 male; M age = 23.3, SD = 7.0)	Anagram problem solving	(1) 25 g glucose, (2) 25 g fructose, (3) Sucralose	3 h	Fructose (*p* = 0.01) and glucose (*p* < 0.01) solved more problems than placebo.
Mohanty and Flint, 2001 [92]	*N* = 70 undergraduate students (22 male; M age = 20.6, SD = 4.3)	Recall of object location task (pattern recognition)	(1) 50 g glucose, (2) 100 mg/kg glucose, (3) 23.7 mg saccharin	8 h	In the emotional condition, more errors were made following glucose ingestion (no *p* values given). In the neutral condition, fewer errors were made following 100 mg/kg glucose compared to placebo. Response time was slower following 50 g glucose. More errors were made in spatial memory retention following glucose ingestion.
Owen et al., 2010 [55]	*N* = 90 undergraduate students (29 male) aged 18–30 (M = 21)	Word list, face recognition test	(1) 25 g glucose, (2) 60 g glucose, (3) Placebo	12 h	Improvement in implicit memory following 60 g glucose (*p* < 0.01). More false alarms after 25 g glucose compared to placebo or 60 g glucose (*p*s = 0.03).
Owen et al., 2012 [56]	*N* = 30 adults aged 18–25 (M = 20)	Word list, serial threes and sevens, Corsi block-tapping, STROOP, simple response time task	(1) 2 h fast and 25 g glucose, (2) 2 h fast and 60 g of glucose, (3) 2 h fast and 0 g glucose, (4) 12 h fast and 25 g glucose, (5) 12 h fast and 60 g glucose, (6) 12 h fast and 0 g glucose	2 h vs. 12 h	60 g glucose increased working memory (speed of recognition (*p* < 0.05) and serial threes (*p* < 0.01)) after fasting. Reaction time impaired with 25 g glucose after fasting.
Owen et al., 2013 [57]	*N* = 24 young adults aged 18–30 (M = 20)	Word list, serial threes and sevens, Corsi block-tapping	(1) 25 g glucose, (2) 60 g glucose, (3) Placebo	12 h	Improved working memory and declarative memory after 25 g and 60 g glucose (*p*s < 0.05). Improved spatial working memory and word recognition after 25 g (*p*s < 0.05) and 60 g (*p*s < 0.01) glucose. No effect of glucose on accuracy. Participants with poor glycaemic control performed better on recall after 25 g glucose (*p* < 0.05). Improved response time after 25 g (*p* < 0.05) and 60 g (*p* < 0.01) of glucose.
Parker and Benton, 1995 [93]	*N* = 100 females (M age = 20.15)	Dichotic listening task, word list, choice reaction time task, auditory word recognition	(1) 75 g glucose, (2) Aspartame and acesulfame K	No	No effect of glucose on recognition Glucose drinkers recalled more when they nominated their right ear than left (*p* < 0.005), whereas placebo drinkers did not. Recall from attended (*p* < 0.013) or unattended (*p* < 0.034) right ear was better when BGL falling rather than rising.
Peters et al., 2020a [30]	*N* = 32 adults (16 younger adults aged 21–30 (8 female, M age = 25.8, SD = 3.2) and 16 older adults aged 55–78 (8 female, M age = 68.6, SD = 6.5))	Serial sevens, Virtual Morris Water Maze	(1) 25 g glucose, (2) 30 mg saccharin	12 h	Older adults performed worse than younger in placebo condition (*p* = 0.02) but not glucose condition. Older adults had poorer glucose regulation (*p* = 0.002) and a greater response to glucose (*p* = 0.006) than younger adults.
Riby et al., 2004 [94]	*N* = 20 older adults (M age = 68.75, SD = 6.0)	Verbal Paired Associates, digit symbol substitution, digit span	(1) 25 g glucose, (2) 38 mg saccharin	Unclear	Greater recall in glucose compared to placebo group (*p* < 0.05); immediate recall most sensitive to glucose (*p* < 0.01).
Riby et al., 2008 [95]	*N* = 33 middle-aged adults (19 females) aged 35–55)	Word list, national adult reading test, digit symbol substitution, letter cancellation, TMT, digit span, category fluency	(1) 25 g glucose, (2) 50 g glucose, (3) Saccharin	2 h	50 g glucose scored higher on word recall accuracy than 25 g or placebo (*p* < 0.001). Good glucose regulators performed better on trail making, except after 50 g glucose (*p* = 0.05). Consumption of “sugar, calories, sweets and drinks” were related to poor glucose control (*p* < 0.05).
Riby et al., 2011 [96]	*N* = 56 adults (25 male) aged 17—80 (M = 34.4, SD = 17.0)	SART	(1) 25 g glucose, (2) Saccharin	3 h	Increased speed of response after glucose (*p* < 0.05). No effect of glucose on accuracy or sustained attention. Good glucose regulators had quicker responses compared to poor regulators (*p* < 0.05).
Scholey et al., 2001 [97]	*N* = 30 adults (11 male) aged 20–30 (M = 27.7)	Serial sevens, word retrieval, word list	(1) 25 g glucose, (2) 30 mg saccharin	9 h	Glucose ingestion led to more responses in serial sevens (*p* < 0.05). BGL fell more during serial sevens (high cognitive load task) regardless of treatment (*p* = 0.009). No effect of glucose on verbal fluency or memory.
Scholey et al., 2009 [98]	*N* = 120 adults (77 female; M age = 21.6, SD = 4.9)	Word recognition (auditory), tracking task	(1) 25 g glucose, (2) 30 mg saccharin	11 h	Improved tracking after glucose ingestion (*p* = 0.045). No effect of glucose on recognition accuracy or reaction time.
Scholey et al., 2014 [58]	*N* = 160 adults aged 18–55	Arithmetic task, STROOP, memory search task, target tracker	(1) 25 g glucose, (2) 60 g glucose, (3) 40 mg caffeine + 60 g glucose, (4) Placebo	12 h	No difference between glucose groups and placebo. Improved scores due to caffeine + glucose.
Serra-Grabulosa et al., 2010 [59]	*N* = 40 students (20 male) aged 18–25 (M = 19.6, SD = 1.7)	Continuous Performance Test—Identical Pairs (CPT-IP)	(1) 75 g glucose, (2) 75 mg caffeine, (3) 40 mg caffeine + 60 g glucose, (4) Placebo	8 h	No effect of treatment on performance.
Smith and Foster, 2008 [99]	*N* = 32 adolescents (12 male) aged 14–17 (mean = 15.6, SD = 0.9) *N* = 10 in glucose second test condition	Modified CVLT-II	(1) 25 g glucose, (2) Aspartame	9.5–10.5 h	No effect of glucose on recall. Treatment x treatment order effect—improved performance for glucose ingestion only when glucose ingested in second session after placebo trial.
Smith et al., 2011b [100]	*N* = 58 adolescent males aged 14–17 (M = 15.5, SD = 1.0)	Modified CVLT-II	(1) 25 g glucose, (2) Aspartame	9.5–11 h	More items recalled after glucose ingestion on 4th (*p* < 0.05) and 5th (*p* < 0.01) trial. Glucose only improved memory in those reporting higher trait anxiety (*p* < 0.05).
Spiers et al., 1998 [101]	*N* = 48 adults (24 male) aged 18–35	Word list, digit span, Corsi block test, TMT, Go-No-Go, Controlled Oral Word Association Test (COWAT), STROOP	(1) 90 g/d sucrose, (2) 15 mg/kg Aspartame, (3) 45 mg/kg Aspartame, (4) 300 mg cellulose (5) (3-month dietary intervention)	No	No effect of treatment on cognitive performance.
Stollery and Christian, 2013 [102]	M = 93 adults (35 male) aged 18–35 (M = 20.7)	Word list recall, spatial location recognition, category verification	(1) 50 g glucose, (2) 75 mg saccharin	Unclear	No effect of glucose on any outcome.
Stollery and Christian, 2015 [103]	*N* = 80 adults (26 male) aged 18–51 (M = 22.4)	PAL	T1: (1) 50 g glucose pre-learning, (2) 50 g glucose post-learning, (3) 75 mg saccharin pre-learning, (4) 75 mg saccharin post-learning T2 (retrieval—1 week): (1) 50 g glucose, (2) 75 mg saccharin	9.5 h	T1: No effect of glucose on performance. T2: Glucose ingestion at retrieval led to improved retention (*p* = 0.016) and lower omission errors (*p* = 0.008).
Stollery and Christian, 2016 [104]	*N* = 31 adults (9 male; glucose M age 22.5, SD = 1.5; placebo M age = 26.5, SD = 4.0)	Object-location binding task	(1) 30 g glucose, (2) 45 mg saccharin	9.5 h	Improved location memory (*p* = 0.029) and object-location binding memory (*p* = 0.006) after glucose ingestion. No effect of glucose on object memory, retrieval time, or errors. Higher BGL associated with better location memory (*p* = 0.027) and binding memory (*p* = 0.012).
Sünram-Lea et al., 2001 [105]	*N* = 60 adults aged 18–28 (M age = 21)	CVLT, Rey–Osterrieth complex figure drawing modified digit Span	(1) 25 g glucose + 9 h fasting, (2) Aspartame + 9 h fasting, (3) 25 g glucose + 2 h fasting after breakfast, (4) Aspartame + 2 h fasting after breakfast, (5) 25 g glucose + 2 h fasting after lunch, (6) Aspartame + 2 h fasting after lunch	9 h or 2 h	Superior performance of glucose on immediate recall interference list (*p* < 0.01), no effect of fasting condition. Short-delay free recall improved after glucose (*p* < 0.001) and after breakfast compared to lunch and 9 h fasted (*ps* < 0.001). Short-delay cued recall, long-delay free recall, and long-delay recognition improved after glucose (*p*s < 0.001); no effect of fasting condition. Long-delay cued recall improved after glucose (*p* < 0.001), and breakfast outperformed lunch (*p* < 0.05). Rey–Osterrieth performance superior after glucose (*p* < 0.005), no effect of condition. No effect of drink or condition on digit span.
Sünram-Lea et al., 2002a [106]	*N* = 80 (18 male) adults aged 18–29 (M age = 20)	Modified CVLT, modified Rey–Osterrieth, complex figure drawing, modified serial sevens	(1) 25 g glucose, (2) Aspartame	2 h	Superior performance of glucose on immediate recall interference list and long-delay recognition (*p* < 0.01). Aspartame improved long-delay cued recall in conditions with no interference (low cognitive load) (*p* < 0.05). Glucose facilitation effect in other short- and long-delay cued and free recall seen only in interference conditions (high cognitive load). Rey–Osterrieth performance superior after glucose (*p* < 0.05). Serial sevens performance superior after glucose (*p* < 0.005).
Sünram-Lea et al., 2002b [50]	*N* = 60 adults (26 male) aged 19–34 (M = 21)	Modified CVLT, serial sevens, ROCF (Delivered at baseline, 15 min, and 24 h after treatment)	(1) 25 g glucose, (2) Aspartame (3) (On day one only)	2 h	Improved performance on delayed free recall after glucose ingestion (*p* < 0.001). Improved 24 h delayed free recall (*p* < 0.001) and recognition (*p* = 0.007) after glucose ingestion. After 30 min, glucose performed better on delayed reproduction of figure (*p* = 0.03). No effect of glucose on immediate free recall or serial sevens performance.
Sünram-Lea et al., 2008 [107]	*N* = 56 young adults aged 18–25 (M = 20)	Word list	(1) 25 g glucose, (2) Aspartame	2 h	Increased recognition responses (recalling words associated with memories or experiences) following glucose ingestion (*p* = 0.04).
Sünram-Lea et al., 2011 [108]	*N* = 30 young adults (6 male) aged 18–25 (M = 20)	Serial threes and sevens, word list	(1) 15 g glucose + 14 saccharin tablets, (2) 25 g glucose +7 saccharin tablets, (3) 50 g glucose +5 saccharin tablets, (4) 60 g glucose + 3 saccharin tablets, (5) 20 saccharin tablets	12 h	Improved spatial working memory (Corsi block task) after 25 g glucose ingestion (*p* < 0.02). Improved immediate free recall after 25 g glucose ingestion (*p* < 0.01). Improved recognition performance after 25 g glucose ingestion (*p* < 0.05). No effect of glucose on numeric working memory (serial threes and sevens) or delayed free recall. Good glycaemic control associated with improved performance after 60 g glucose. Poor glycaemic control associated with improved performance after 15 g glucose.
van der Zwaluw and et al., 2014 [61]	*N* = 43 older adults (16 male; M age = 77.7, SD = 5.6)	RAVLT, PAL, story recall, verbal fluency, digit span, STROOP, Test for Attentional Performance (TAP)	(1) 50 g glucose, (2) 100 g sucrose, (3) Placebo	10–12 h	Improved attention, working memory, and information processing after sucrose compared to placebo (*p* = 0.04). Improved tap flexibility.
Walk et al., 2017 [60]	*N* = 113 children aged 9–10	Erikson flanker task	(1) Sucrose, (2) Maltodextrin, (3) Carbohydrate blend (68% isomaltulose, 9% maltodextrin, 13% Fibersol-2), (4) Sucralose	10 h	No effect of treatment on cognitive performance. When glycaemic regulation was adjusted for, only Maltodextrin had improved reaction time (*p* = 0.044).
Winder and Borrill, 1998 [109]	104 adults (52 male) aged 18–55 (mean = 29.2, SD = 9.23)	Name–face association task, selective reminding task	(1) 50 g glucose with oxygen, (2) 50 g glucose with air, (3) 4 g aspartame with oxygen, (4) 4 g aspartame with air	No	No effect of glucose on performance.

M = mean; RAVLT = Rey Auditory Verbal Learning Memory Test; WCST = Wisconsin Card Sorting Test; WAIS = Wechsler Adult Intelligence Scale; TMT = trail making tests; RIPT = rapid information processing task; SART = sustained attention to response task; CVLT = California Verbal Learning Test; CDR = Cognitive Drug Research; ROCF = Rey–Osterrieth Complex Figure; PAL = paired-associate learning; BGL = blood glucose level.

### 3.8. Blood Glucose Regulation

Glucose regulation was calculated using one or both of two strategies: measuring recovery from evoked glucose levels or calculating the area under the curve of evoked glucose levels. A total of 24 intervention studies investigated the role of glycaemic control on performance, with 20 of these finding a significant effect of glucose regulation on performance. These studies analysed blood glucose levels (BGL) over a period of 40 to 180 min. The majority of studies found that poor glucose regulation was associated with poorer performance, and falling BGL was associated with improved performance [31,56,61,62,67,71,78,79,84,86,89,93,94,95,96,99]. Three studies found that glucose administration could attenuate the cognitive deficits seen in younger [57] and older [62,110] participants with poor glycaemic control. Differences in age were also noted as older participants had poorer glucose regulation [77,110]. Craft et al. [77] found that older men with good recovery and younger men with poor recovery had comparable BGLs and benefitted similarly from 50 g of glucose, whereas younger men with good glucose recovery had impaired performance. Owen et al. [57] also found that glucose consumption (25 g) could impair memory in those with superior glucoregulation. However, Sunram-Lea et al. [108] found that in a sample of young adults, good regulators benefited from much higher doses (60 g) of glucose, whereas poor regulators benefited from lower doses (15 g). Sunram-Lea et al. [106] found a positive correlation between BGL and a number of memory performance measures which were not present when only the glucose condition was analysed, suggesting that 25 g glucose consumption may have breached the upper threshold of optimal BGL in this young adult sample. Four studies found no association between AUC or changes in BGL with cognitive performance [63,80,104,107]. These studies analysed changes in BGL over 30–53 min, which may not have been long enough to identify significant changes in BGL.

### 3.9. Meta-Analysis for Word List Recall Studies

Due to the variety of cognitive outcomes, treatments, and measures used in the studies, only 16 papers contained sufficient data for word list recall and were included in the meta-analysis; 13 measured immediate free recall, and 12 measured delayed free recall (see Figure 2 and Figure 3). Analysis was conducted using a random effects model with standard mean difference (SMD) as the effect measure. Overall, there was a significant effect of glucose on immediate free recall (SMD = 0.22, 95% CI = 0.08 to 0.36, *p* = 0.002) but not delayed free recall (SMD = 0.24, 95% CI = −0.04 to 0.51, *p* = 0.10). The Knapp–Hartung correction has been proposed to provide a more conservative and accurate result [111]. When meta-analysis was conducted using SPSS with a Hartung–Knapp correction, results were similar for immediate free recall (95% CI = 0.07, 0.38, *p* = 0.006) and delayed free recall (95% CI = −0.19, 0.72, *p* = 0.233). Subgroup analysis showed a significant effect of glucose only in the parallel design studies for immediate free recall (SMD = 0.34, 95% CI = 0.15 to 0.52, *p* < 0.001). This could be due to possible bias in the randomisation or sampling techniques used for parallel trials. Results may also overestimate the impacts of sugar as the majority of papers that failed to include data for free recall reported no effect of sugar on these measures or failed to report the results completely [33,67,79,82,97,101,102].

### 3.10. Heterogeneity of Reported Outcomes and Study Bias

In the meta-analysis of immediate free recall, overall heterogeneity was low (I^2^ = 1%), reflecting low heterogeneity in both the crossover design trials (χ^2^ = 8.24, *p* = 0.51, I^2^ = 0%) and in the parallel design trial (χ^2^ = 8.73, *p* = 0.56, I^2^ = 0%). Substantial heterogeneity was detected in the delayed free recall studies (I^2^ = 70%), both in the crossover trials (χ^2^ = 20.26, *p* = 0.04, I^2^ = 46%) and the parallel design trials (χ^2^ = 32.69, *p* < 0.001, I^2^ = 88%). This may reflect the variance in time periods that were considered “delayed recall” in each study. Findings could also be caused by randomisation issues or population characteristics such as age and gender, which were often poorly described. Based on these findings, the results of the analysis for delayed immediate recall should be interpreted with caution. A visual inspection of the funnel plot for immediate free recall (see Figure 4) saw little evidence of asymmetry. Egger’s test of small study bias showed no risk of bias for immediate free recall (*p* = 0.963) or delayed free recall (*p* = 0.085).

### 3.11. Sensitivity Analysis

The influence (leave-one-out) analysis conducted on immediate free recall showed no significant change in heterogeneity, mean differences, or significance levels (*p* values). When investigating the delayed free recall meta-analysis, removal of Smith and Foster [99] reduced I2 to 0% for crossover designs but had no impact on significance. Removal of Sunram-Lea et al., 2002a [106], however, led to an overall effect of glucose (SMD = 0.29, 95% CI = 0.03, 0.56, *p* < 0.03) (see Appendix A).

### 3.12. Strengths and Limitations

A strength of the current study is the inclusion of multiple study designs and sugar types. This allows a broad and multifaceted understanding of the effects of sugars (both short- and long-term effects) and the characteristics and volume of previously conducted studies. The inclusion of natural sugars was deemed important to develop a more reliable conclusion from observational studies given that food frequency questionnaires often fail to differentiate between natural fructose and added fructose. Previous studies have included natural fructose with other sugar types in their analyses of the impact of sugars, which could lead to conflicting or unreliable results if natural fructose from fruits and juices proves to have contrasting impacts on cognitive function or health.

A potential confounder in almost all the experimental studies was the use of artificial sweeteners as a control substance. Artificial sweeteners have previously been linked to cognitive and behavioural changes in human and animal models [112,113]. A large cohort study found that artificially sweetened drinks were associated with a higher risk of stroke and dementia [114]. Animal studies have observed impaired glucose tolerance and cognitive dysfunction after the consumption of artificial sweeteners [115,116,117] as well as increased anxiety that was transmitted to subsequent generations [118]. The impact of these artificial sugars on behaviour and cognition has the potential to obscure or exaggerate the real effects of sugar.

The extensive array of cognitive assessments used across the studies made comparisons between them difficult. Tests measured diverse domains of function, some including numerous related subdomains. For example, measures of memory included spatial, verbal, visual, and working memory. Each of these tests has the potential to address unique neural correlates, which may or may not be sensitive to BGL or long-term, sugar-induced inflammatory processes. Further understanding of the neurophysiology associated with cognitive domains and assessments, and improved knowledge of the specific actions of sugar, may highlight the most optimal cognitive tests to use to assess these relationships. Further in-depth analysis of findings from experimental papers may also have uncovered unique patterns of impairment or enhancement, such as Parker and Benton’s [93] findings that suggested lower BGLs were associated with improved attention, while higher BGLs were associated with improvements in memory. While interesting, these detailed investigations are outside the scope of this review.

Research investigating individual nutrients typically requires adjustment in order to separate the effects of an individual dietary component from the effects of the total diet [119]. An individual who consumes a large amount of food will have a larger intake of any individual dietary constituent, and participants with a greater body mass will likely require a greater amount of nutrients to see similar effects to those in a much smaller individual. A number of strategies can be conducted in order to control for total energy intake. For observational studies, all but one [47] paper adjusted for total energy intake by normalising sugar for total daily consumption or included total energy consumption into statistical modelling. Randomised control trials utilised the same amount of glucose for each participant, independent of body mass, which may have led to confounding of results.

## 4. Discussion

### 4.1. Observational Research

The 12 cross-sectional and cohort studies identified in the review found that chronic overconsumption of sugar negatively correlated with measures of global cognitive function, executive function, and memory. These results are similar to the findings of numerous animal studies that investigated the short- and long-term impacts of sugar consumption, finding alterations in neurophysiology and related impairment in cognitive function, specifically memory [12,25,26,120,121]. The present findings also identified two papers of mother–infant pairs that showed correlations between maternal dietary sugar intake and impairment in infant cognitive function. These have also been replicated in several animal studies, showing cognitive deficits and hippocampal alterations in offspring exposed to maternal high-sugar diets [122,123,124].

The current findings produced only a small number of large-scale or longitudinal investigations of the impact of dietary sugar. However, these align with those of several population-based studies that were not contained in our analysis as they included secondary findings from larger national studies. Subsamples of the National Health and Nutrition Examination Survey (NHANES) (USA) [28,125], the Chinese Longitudinal Healthy Longevity Survey (CHLS) [126], China Health and Nutrition Survey (CHNS) [127], the Seguimiento Universidad de Navarra (SUN) cohort (Spain) [128], the Framingham Heart Study (USA) [113,129], and the Kids in Taiwan: National Longitudinal Study of Child Development and Care (KIT) study [130] also found associations between increased dietary sugar and SSB consumption and cognitive dysfunction or increased risk of cognitive impairment or dementia. On the other hand, a CLHS subsample of 7572 older Chinese adults found no relationship [131], and a subsample of 1809 adolescents from the NHANES 1988–1994 found improved arithmetic, reading, and digit span associated with increased consumption of SSBs [132]. These inconsistent results highlight the potential for sample age, cultural differences, or sampling and analytic methods to impact study outcomes in observational study designs and thus indicate the need for additional, consistent, and rigorous research.

Considering the small number of studies conducted, additional, large-scale cohort studies including a variety of measures of cognitive function are required to better investigate relationships and potential confounders. Cohort studies are more reliable than cross-sectional studies as they are able to assess multiple exposures over time and better investigate confounding or influential factors. While all included studies controlled for several covariates, such as BMI, education, and age, there are likely to be a number of other factors that have not been recognised or accounted for. Cross-sectional and cohort studies also commonly rely on food frequency questionnaires to collect data on dietary consumption. These rely on participant recall, adherence, and a consistent interpretation of serving sizes, leading to increased chance of error. Studies also reported an inability to determine the origin of fructose consumption (from added or natural sources), which could interfere with the interpretation of results.

### 4.2. Experimental Research

The review identified 65 experimental studies that investigated the immediate and short-term impact of sugar on cognition. The majority of these studies found a beneficial effect of sugar consumption on one or more measures of cognitive function, eliciting the glucose facilitation effect in tests of memory, attention, and processing speed. The outcomes of these studies were largely dependent on individual differences in fasting status, baseline BGL, glucose regulation, and the cognitive effort required of the task. These findings support the notion that BGLs must be tightly regulated and maintained at an ideal, individual level for optimal cognitive performance. Due to the large variability seen in individual glucoregulation and performance, counterbalanced cross-over trials would be the most appropriate method for conducting future RCTs.

The majority of studies used a procedure of fasting participants for up to 12 h before testing. This makes it difficult to determine if the tests measured the impact of sugar administration or the impact of low BGL on cognitive performance. Other substances showed similar or better performance after consumption. Jones [32] identified superior memory after protein ingestion and enhanced response times following fat ingestion. These findings indicate that sugar may not be the ideal substance to improve cognitive function in those with low BGL or impaired glucoregulation, particularly if long-term consumption of sugars is a risk factor for impaired glucoregulation and reduced insulin sensitivity.

### 4.3. Sugar Types

Experimental studies almost exclusively used glucose. This makes it difficult to determine the short-term impact of different sugar types on cognitive function. Glucose was measured in two cross-sectional studies and one cohort study. One cross-sectional study of 1209 older adults and a cohort study of 737 mother–infant pairs found impaired global cognitive function associated with glucose. However, a cross-sectional study of 487 children aged 6–8 years found no relationship. This same pattern was observed in sucrose, with one additional cohort study of mother–infant pairs finding impairment in global cognitive function and memory. This may reflect a window of vulnerability for prenatal women and young infants. Added sugars were seen to impair cognitive function in adults and adolescents but not in children aged 7–9 years. The lack of association in children may be due to the reduced time period of consumption, or alternatively, it may be due to ceiling effects and a reduced overall consumption of sugars by this young age group that is being cared for by parents or other caregivers.

One inconsistent finding was observed in evaluations of fructose. Natural fructose is consumed from fruits and 100% fruit juice. However, the main source of fructose in the Western diet is sucrose (made from equal parts glucose and fructose) added to foods and beverages and high fructose corn syrup found in SSBs. Impairments in cognitive function were found in all age groups when investigating SSBs. Four studies measured intake of fructose from fruits and sugars. Three of these (in children, mother–infant pairs, and older adults) found enhancements in global cognitive function associated with consumption [14,45,47]. One study of adults found no association [48]. In contrast to this, consumption of added fructose was associated with lower cognitive function [48]. Total dietary fructose was associated with improved function in a study of children [43] and reduced performance in older adults [14]. These findings could be due to differences in diet, with young children more likely to consume fructose in the form of fruits and juice.

## 5. Conclusions

Studies of diet and dietary macronutrients are abundant; however, studies investigating the independent impact of sugar on health and cognition are surprisingly lacking. The bulk of studies examining the impact of sugar on domains of cognitive function are short-term experiments that highlight the glucose facilitation effect in fasted participants. These studies have identified that tightly regulated blood glucose and sugar consumption is required for optimal cognitive function. However, this optimal level is defined by individual physiology, age, and lifestyle factors. The benefits observed may also be induced by nonsugar macronutrients.

Contrasting short- and long-term effects of sugar on cognitive function have been observed. Acute facilitation effects of sugar on cognitive function have been observed for glucose after fasting for 2 to 12 h. In contrast, negative impacts of sugar on cognitive function seem to be associated with excessive, long-term, and prenatal consumption, particularly from high fructose corn syrup found in SSBs. Natural fructose from fruits and fruit juices was associated with improved cognitive function. However, it is important to consider the other constituents of fructose-containing foods. For example, fruits contain fibre and other vitamins and minerals that may have beneficial impacts that counteract the potential detrimental effects of sugars. In contrast, foods high in added fructose may be highly processed, lacking in beneficial nutrients, or high in potentially disadvantageous additives. More research into natural sugars should be conducted to untangle the potential benefits of fibre from the effects of sugars and to determine if a recommended daily intake range is necessary. Longer-term RCTs (of weeks or months) investigating the impact of individual sugar types and additional cohort studies that address potential confounders (sugar source, other nutrient intake, exercise, glucose regulation, medications, or psychiatric disorders) are needed to provide an improved understanding of the potential short- and long-term impact of sugar on cognition.

## Figures and Tables

**Figure 1 nutrients-16-00075-f001:**
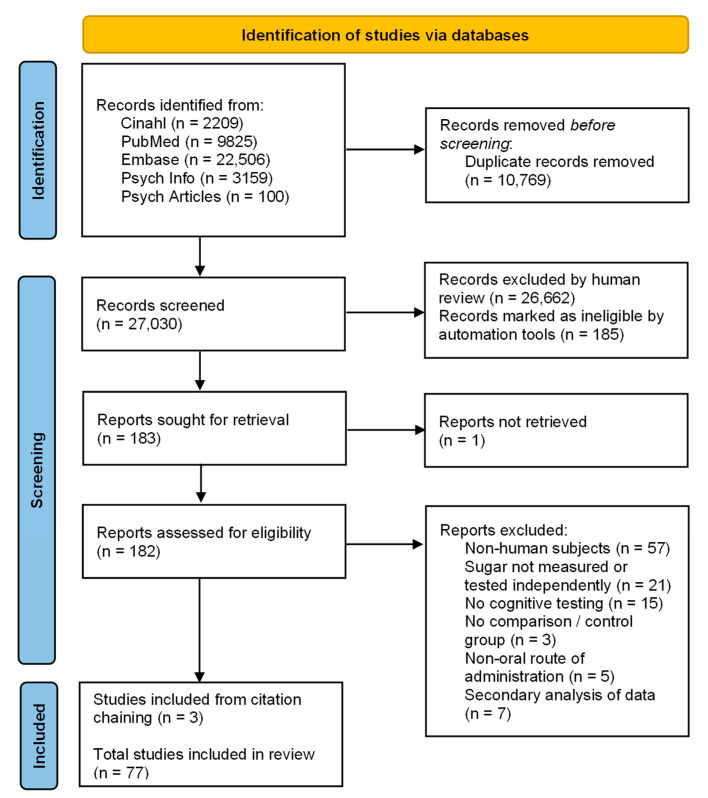
PRISMA flow diagram [37].

**Figure 2 nutrients-16-00075-f002:**
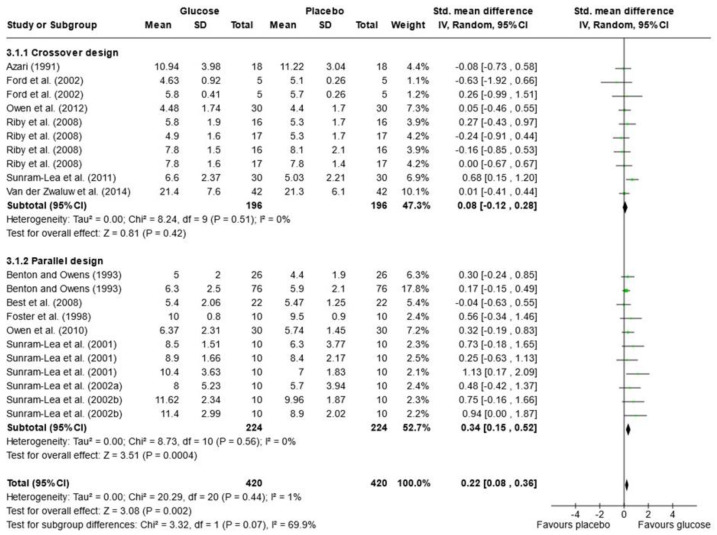
Random-effects meta-analysis of standard mean differences (SMD) and 95% confidence intervals (CIs) for studies that obtained data for immediate free recall [50,52,55,56,61,63,66,69,81,95,105,106,108]. Data markers represent the SMD, and their size represents the weight assigned to each study. Horizontal bars represent the 95% CI. The black diamonds represent pooled summary analyses. The P symbol represents *p* significance.

**Figure 3 nutrients-16-00075-f003:**
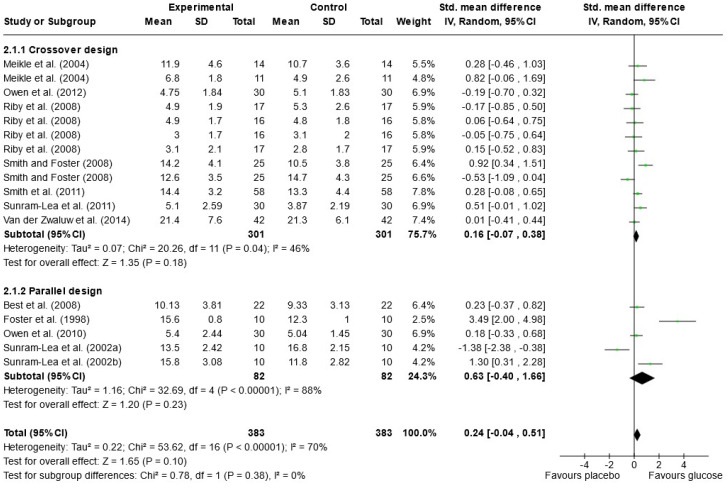
Random-effects meta-analysis of standard mean differences (SMD) and 95% confidence intervals (CIs) for studies that obtained data for delayed free recall [50,52,53,55,56,61,69,95,99,100,106,108]. Data markers represent the SMD, and their size represents the weight assigned to each study. Horizontal bars represent the 95% CI. The black diamonds represent pooled summary analyses.

**Figure 4 nutrients-16-00075-f004:**
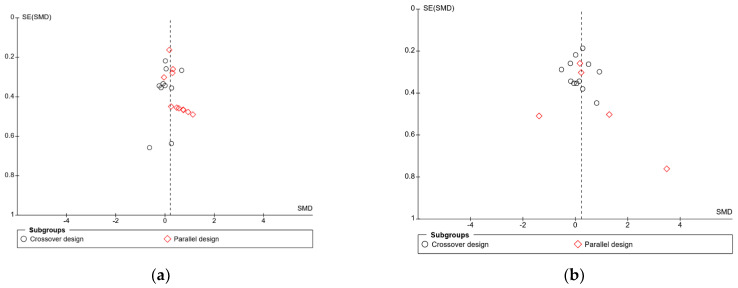
Funnel plots for (**a**) immediate free recall and (**b**) delayed free recall.

**Table 1 nutrients-16-00075-t001:** Observational study characteristics.

Author/s	Population	Cognitive Measures	Dietary Measures	Sugar Type	Major Findings
Al-Sabah et al., 2020 ^a^ [44]	*N* = 1370 adolescents aged 11–16	Raven’s Standard Progressive Matrices (SPM)	FFQ	Sugary drinks (frequency of consumption)	Inverse relationship between sugary drink consumption and cognitive performance (*p* < 0.001). Sugary drink consumption a main predictor of cognitive function (*p* < 0.001).
Baym et al., 2014 ^a^ [40]	*N* = 52 children aged 7–9	Memory tasks (paired associated images)	Youth–adolescent FFQ (YAQ)	Total sugar, added sugar (normalised by total daily kcal consumption)	Total and added sugars had no relationship to memory. Performance impacted negatively by saturated fatty acids and positively by omega-3 fatty acids (*p*s < 0.05).
Berger et al., 2020 ^b^ [46]	*N* = 88 mother–infant pairs (mother M age = 68.1, SD = 6.7)	Bayley-III scales of infant development	FFQ at 1 and 6 months postnatal	Fructose, SSB (incl. juice), total sugar, added sugar (adjusted for kcal per day)	Maternal fructose (*p* < 0.01) and SSB (*p* = 0.02) consumption at 1 month postnatal negatively associated with infant cognitive development at 24 months. Infant cognition lower in infants of obese mothers (*p* ≤ 0.001). No effect of maternal intake at 6 months.
Chong et al., 2019 ^a^ [14]	*N* = 1209 older adults aged 60 years and over (M = 68.1, SD = 5.6)	RAVLT, MMSE, MoCA, digit symbol (processing speed test, not described), VR	Dietary history questionnaire (1 week recall)	Fructose, glucose, total sugar, added sugar, SSB, sugar from cakes and deserts, adjusted for daily calorie intake)	Mild cognitive impairment had higher consumption of fructose (*p* = 0.004) and glucose (*p* = 0.032). MMSE scores lower in the higher percentile of total and free sugar intake (*p*s < 0.001). Risk of cognitive impairment increased 3.3-, 3.3-, and 3.6-fold in highest percentile of sucrose, total sugar, and free sugar, respectively (*p*s < 0.001). Risk of cognitive impairment increased by 3.7- and 1.8-fold for SSB and sugar from cakes and deserts, respectively (*p*s < 0.001). Reduction in risk of cognitive impairment by 35% in highest percentile of fruit consumption (*p* < 0.05).
Cohen et al., 2018 ^b^ [47]	*N* = 1234, mother–child pairs (tested during pregnancy and early childhood)	Peabody Picture Vocabulary Test (PPVT-III), WRAVMA, KBIT-II, WRAML	FFQ	Maternal (prenatal) SSB consumption, child SSB consumption (not adjusted for energy intake)	Maternal sucrose ingestion inversely associated with nonverbal KBIT-II (*p* = 0.03) and visual memory (*p* = 0.01) in mid-childhood. Maternal SSB ingestion inversely associated with nonverbal KBIT-II in mid-childhood (*p* = 0.03). Maternal diet soda consumption associated with lower WRAVMA in early childhood (*p* = 0.03) and verbal KBIT-II in mid-childhood (*p* < 0.001). Early childhood SSB consumption inversely associated with verbal KBIT-II in mid-childhood (*p* = 0.01). Fructose (*p* = 0.005) and fruit (*p* = 0.03) positively associated with PPVT-II in early childhood.
Gui et al., 2021 ^a^ [41]	*N* = 6387 children (3410 male) aged 6–12 (M = 8.6, SD = 1.5)	BRIEF	FFQ	HFCS from SSB (frequency of consumption)	Associated with poor performance on executive function and high risk of executive dysfunction (*p*s < 0.0001).
Hassevoort et al., 2020 ^a^ [39]	*N* = 54 children (31 female) aged 8–12 (M = 9.1, SD = 0.8)	TTCT—Verbal form A	3-day FFQ	Added sugar (normalised to intake per 1000 kcal)	Inversely associated with fluency, originality, and overall TTCT score (*p* < 0.01).
Lester et al., 1982 ^a^ [38]	*N* = 184 children (100 male) aged 5–16	WISC-R, WIPPSI, WRAT	FFQ—24-h recall	Refined carbohydrates (adjusted for total calories)	Negative relationship with all aspects of cognition (full-scale IQ (*p* = 0.001), performance IQ (*p* = 0.025), verbal IQ (*p* = 0.005), math (*p* = 0.005), and reading (*p* = 0.025)) other than spelling. Ratio of refined carbohydrates to total food calories negatively correlated with full-scale IQ (*p* < 0.015).
Naveed et al., 2020 ^a^ [43]	*N* = 487 children (250 male) aged 6–8	Raven’s Coloured Progressive Matrices (RCPM)	FFQ	Fructose, sucrose, glucose (adjusted for daily energy intake)	Increased fructose intake associated with higher fluid intelligence scores in all children (*p* = 0.002) and boys (*p* ≤ 0.001) but not girls alone; effect disappeared in all children and reduced in boys when fruits and berries were accounted for. No effect of glucose or sucrose
Øverby et al., 2013 ^a^ [45]	*N* = 482 students (236 male, M age = 14.6)	Self-reported schooling difficulties (maths, reading, and writing)	FFQ	SSB, junk food (frequency of consumption)	Higher intake of SSB (*p* = 0.04) and junk food (*p* ≤ 0.001) associated with increased odds of self-reported math difficulties. Fruit intake associated with lower odds of math difficulties. No relationship with reading and writing difficulties.
Ye et al., 2011 ^c^ [48]	*N* = 737 adults aged 45–75	MMSE, word list learning, digit span, clock drawing and figure copying (visual–spatial), STROOP, verbal fluency test	FFQ (12-month estimate)	Total sugar, added sugar, SSB (adjusted for total energy intake)	Increased sucrose (*p* = 0.014), glucose (*p* = 0.032), SSB (*p* = 0.005), and added fructose (*p* = 0.028), but not natural fructose, associated with lower MMSE. Total sugars inversely correlated with letter fluency (*p* < 0.05), recognition and recall (*p*s < 0.05), memory (*p* = 0.01), and MMSE (*p* = 0.02). Added sugars inversely associated with letter fluency (*p* < 0.05), long-term recall (*p* < 0.05), and MMSE (*p* = 0.005).
Zhang et al., 2022 ^a^ [42]	*N* = 1231 adolescents aged 13–18 (M = 15.77, SD = 1.7)	Modified Erikson flanker task, 1-back and 2-back tasks, more-odd shifting task	FFQ	SSB (frequency of consumption)	Drinking SSBs ≥2 times per week had worse performance for inhibition, working memory, and cognitive flexibility than no SSBs (*p*s < 0.001).

Note: ^a^ = Cross-sectional study design; ^b^ = prospective cohort study design; ^c^ = retrospective cohort study; M = mean; FFQ = food frequency questionnaire; MMSE = Mini Mental State Exam; SSB = sugar-sweetened beverages; RAVLT = Rey Auditory Verbal Learning Memory Test; MoCA = Montreal Cognitive Assessment; VR = Visual Reproduction Test; WRAVMA = Wide-Range Assessment of Visual Motor Abilities; KBIT-II = Kaufman Brief Intelligence Test; WRAML = Wide-Range Assessment of Memory Learning; BRIEF = Parent-Rated Behavioural Rating Inventory of Executive Function; HFCS = high-fructose corn syrup; TTCT = Torrance Test of Creative Thinking; WISC-R = Wechsler Intelligence Scale for Children; WIPPSI = Wechsler Preschool and Primary Scale of Intelligence; WRAT = Wide-Range Achievement Test.

**Table 3 nutrients-16-00075-t003:** Summary of findings of added * sugars on cognitive outcome measures.

Cognitive Domain	Study Type	Enhanced N (%)	Impaired N (%)	Mixed (Enhanced and Impaired) N (%)	No Effect/Not Reported
Global cognitive function	Cross-sectional	0 (0.0)	7 (77.8)	0 (0.0)	2 (22.2)
	cohort	0 (0.0)	7 (100.0)	0 (0.0)	0 (0.0)
Memory	Intervention	24 (42.1)	2 (3.5)	6 (10.5)	25 (43.9)
	cross-sectional	0 (0.0)	2 (28.6)	0 (0.0)	5 (71.4)
	cohort	0 (0.0)	2 (33.3)	0 (0.0)	4 (66.7)
Coordination	Intervention	4 (50.0)	0 (0.0)	0 (0.0)	4 (50.0)
	cohort	0 (0.0)	0 (0.0)	0 (0.0)	2 (100.0)
Attention	Intervention	14 (29.2)	4 (8.3)	1 (2.1)	29 (60.4)
	cross-sectional	0 (0.0)	1 (20.0)	0 (0.0)	4 (80.0)
	cohort	0 (0.0)	0 (0.0)	0 (0.0)	4 (100.0)
Perception	Intervention	6 (30.0)	0 (0.0)	0 (0.0)	14 (70.0)
	cohort	0 (0.0)	0 (0.0)	0 (0.0)	6 (100.0)
Processing speed	Intervention	15 (29.4)	5 (9.8)	1 (2.0)	30 (58.8)
	cross-sectional	0 (0.0)	0 (0.0)	0 (0.0)	5 (100.0)
	cohort	0 (0.0)	0 (0.0)	0 (0.0)	4 (100.0)
Executive function	Intervention	6 (25.0)	0 (0.0)	1 (4.2)	17 (70.8)
	cross-sectional	0 (0.0)	3 (100.0)	0 (0.0)	0 (0.0)
	cohort	0 (0.0)	1 (25.0)	0 (0.0)	3 (75.0)

* To avoid confusion and any potential bias attributable to added vs. natural fructose, we have include only added sugars in this table (sucrose, glucose, SSBs, and refined carbohydrates).

## Data Availability

Data are contained within the articles.

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
