# Peer review of "The Impact of Free and Added Sugars on Cognitive Function: A Systematic Review and Meta-Analysis"

_nutrients, 2023, doi:10.3390/nu16010075_

Round 1
Reviewer 1 Report
Comments and Suggestions for Authors
Comments to the Author
In this paper, the authors aimed to investigate and report the current evidence of the effects of sugars on cognitive function by performing a systematic review and meta-analysis. This study is some interesting and the results may be useful. However, I have some queries and suggestions that are noted below.
1- This review was not prospectively registered such as PROSPERO.
2- Meta-regression should be added and described in manuscript. This concern needs to be clarified.
3- Systematic review strategy needed to be clarified. The author should update the lasted published literature to the results of this manuscript.
4- Lack of sensitive analysis.
5- How to explain super high heterogeneity? (I2=86%)
6- Please add PRISMA 2020 flow chart.
7- How about the HK correction method for small number of studies in Figure 3?
8- There are some grammatical errors in this paper.
Reviewer 2 Report
Comments and Suggestions for Authors
The authors have carried out a systematic review and meta-analysis of the relationship between dietary sugar and cognitive status. This is of much potential value for two reasons. First, it adds to the weight of evidence that sugar is harmful to health. Second, it contributes valuable information regarding the causes of impaired cognitive function and how this problem may be prevented, at least to some degree.
The authors are to be commended for carrying out a very thorough analysis of the currently available publications. This paper will likely be of much value for researchers in the field.
Table 1 includes both cross-sectional studies and cohort studies. However, it is not clear which studies are which. The authors need to add some brief notes to the table to make it clear which studies are of which type.
P 20, lines 270-283, a paragraph is repeated.
P 24, lines 416-417, to avoid any confusion add “on the one hand” and “on the other hand” at appropriate places in the sentence.
P25, line 426, perhaps add the suggestion that a wide variety of measures of cognitive function should be included in the proposed future studies. It might also be useful to point out the greater reliability of cohort studies as compared to cross-sectional studies.
Comments on the Quality of English LanguageExcellent (just needs a few minor edits)
Round 2
Reviewer 1 Report
Comments and Suggestions for Authors
No further comments.